# Conceptual Model of Differentiated-Instruction (DI) Based on Teachers' Experiences in Indonesia

Enung Hasanah [1,*], Suyatno Suyatno [1], Ika Maryani [2], M Ikhwan Al Badar [3], Yanti Fitria [4] and Linda Patmasari [1]

1   Department of Education Management, Universitas Ahmad Dahlan, Yogyakarta 55161, Indonesia
2   Elementary School Teacher Education, Universitas Ahmad Dahlan, Yogyakarta 55161, Indonesia
3   Japanese Language Education, Muhammadiyah Yogyakarta University, Yogyakarta 55183, Indonesia
4   Primary Education, Padang State University, Padang 25171, Indonesia
*   Correspondence: enung.hasanah@mp.uad.ac.id

**Abstract:** Ideally, learning should help students optimally develop through various activities that enable students to pay attention to their well-being. Some teachers in Indonesia have participated in various professional development programs based on developing a progressive educational philosophy emphasizing the importance of differentiated learning to create Indonesia's graduate profile, namely the *Pancasila Student Profile*. Implementing differentiated learning in Indonesia to realize the outcome of the Pancasila Student Profile is a novelty. Therefore, many teachers are still bewildered about the best practice to realize it. This study aims to construct a conceptual model of differentiated instruction based on the experiences of junior high school teachers in Indonesia through phenomenological studies. The number of participants in this study was 12 people selected through the purposive sampling method. Data collection is enacted through in-depth individual interviews. This study indicates that the conceptual model of differentiated learning is learning that provides opportunities for students to walk on their tracks; it also provides learning that emphasizes students' physical and mental welfare and safety in the learning process. Teachers who comprehend and understand the philosophy of progressive education would possibly realize the concept of differentiated learning, which places students as empowered learners.

**Keywords:** differentiated learning; secondary school teacher; true-learner; student well-being

## 1. Introduction

Indonesia [1,2] is a multicultural country [3] with thousands of islands, resources, and vibrant culture [4]. Indonesia's diverse natural and cultural wealth brings about the emergence of distinctions in students' socio-cultural life, language, religion, and economic status. In addition, the differences in the life backgrounds of students in Indonesia are one of the causes for the emergence of differences in their potential, needs, and learning abilities [5]. The diversity of students' potential and learning abilities is a national asset that needs to be developed optimally to realize the ideal next generation of the nation, in keeping with the ideological values of the Indonesian nation.

In order to develop the quality of education for the younger generation, the Indonesian government has made various policy breakthroughs in the field of education. The government set an ideal graduate distinction for Indonesian education, which they labeled as 'Student Profiles.' Student Profiles generally consist of the following characteristics: faith, i.e., God-fearing and noble character; independence; cooperation; diversity; critical reasoning; and creativity [6]. The profile is a concept that describes the ideal picture of students in Indonesia with the characteristics of being a lifelong student who is competent and holds character according to the values of Pancasila [7]. Pancasila is the foundation of the Indonesian state [8].

The program of Pancasila Student Profile development is a part of the efforts made by the Ministry of Education and Culture to strengthen the quality and relevance of education centered on student development [9]. This policy was brought up based on the recognition that every child has a natural potential individually [10], who is rightful to fair health, education, and protection [11,12]. However, millions of children are not given a fair chance due to differences in nationality, gender, or the circumstances in which they were born [13].

Injustice toward children [14,15] also often occurs in classrooms [16]. Students with distinctive essential potential [17] and who have different cultural backgrounds [18], socioeconomic histories [19], needs [20], capabilities [21,22], and skills [23] must pass the same test in the same way [24]. Injustice felt by children is more extensive when children from 3T (frontier, outermost, and least developed) regions [25] must master the same competencies as children from cities that relatively have more and easier accessible educational facilities [26]. The difference in the potential and technology accessibility between students in the 3T regions and in big cities widens the gap of quality in education between regions in Indonesia. Therefore, there needs to be an effort to equalize the quality of education in the 3T regions, taking into account the diversity of children's socio-cultural backgrounds in order to create a prosperous students' Pancasila Student Profile (student well-being).

Ideally, the competencies and characters described in the Pancasila Student Profile are built in everyday life and brought to life in each student through school culture, intracurricular learning, projects, and extracurricular activities [27]. However, this is not an easy task to realize, especially for middle school teachers. To build a Pancasila Student Profile, the government of Indonesia has launched a new paradigm of a learning program. In the learning promoted by the Ministry of Education, there is a concept of differentiated learning or differentiated instruction [28]. With the concept of differentiated learning, the learning process in the classroom can be a medium for teachers to help students become able to have the character of Pancasila Student Profile [29], through a fun process [30], according to a student's needs, and accommodate the background diversity of students. Therefore, in the process of preparing lesson plans, teachers are expected to be able to identify various factors that influence the success of learning.

Teachers' concern and knowledge about the factors that influence student learning outcomes can improve the quality of educational services that they provide to their students. Teachers can develop an ethic of perseverance through the experience of formative assessment that they carry out, which is closely related to the science identity that the students are studying. However, teachers' understanding of what enables students to persevere through challenging assignments shows apparent differences in the emotional and instructional support techniques teachers provide in classrooms [31]. Thus, teachers need to master various socio-emotional teaching techniques [32] to carry out differentiated learning.

Differentiated instruction is an option to carry out meaningful learning for students, develop students' natural potential, create stability in the learning process, and recognize differences in one's potential [33]. Differentiated instruction uses diagnostic assessment data to modify the curriculum and teaching strategies to respond to differences in readiness, interests, and learning profiles [34] so that students are successful in learning. The points above are essential to be noticed, as differentiated learning based on group dynamics has helped students to be more connected and competent. This resulted in more success in both curriculum aspects and social-emotional tasks they will face in their life [35]. The practice of differentiated instruction is positively related to the prosperity of school, social inclusion, and academic self-concept [36].

Various research has shown that many factors can affect the level of student learning success, including factors related to students, factors related to teachers, factors related to the learning environment, and factors related to the social environment [37–40]. Internal factors directly related to students become the most critical factors affecting learning effectiveness. These factors include productive motivation, attitudes, anxiety, empathy, internal barriers, personality [41], and student independence [42,43]. For this reason, learning planning needs to be appropriately set so that its implementation corresponds to

the lives and livelihoods of children in harmony with the children's world [33]. The word in harmony with the children's world means according to children's level of development because education belongs to the children, not to parents or teachers [44]. In this context, differentiated learning can be a solution so that the education experienced by children follows their needs and potential.

Differentiated learning is proven to be able to produce optimal learning achievement for students with special needs [45], those deemed 'gifted' [46,47], as well as those who have limited use of language [48]. Hence, many believe in the efficacy of differentiated learning strategies as a solution to the educational gap in various parts of Indonesia with various cultural backgrounds. Although many parties agree on the importance of conducting differentiated learning [49–52]; however, not many studies have explored effective differentiated learning models in remote areas in Indonesia. This study explores a conceptual model of differentiated learning based on teachers' experiences in remote areas.

## 2. Materials and Methods

### 2.1. Study Design

This research aims to explore the experience of Indonesian teachers in conducting differentiated learning. Based on the literature study by Neubauer, Witkop, and Varpio (2019), as well as Williams (2021) [53,54], it is shown that the suitable methodology to be used in this study is phenomenology. Phenomenology is a research method used to explore phenomena based on the experience of participants who directly experience the phenomenon being studied.

### 2.2. Study Participants

The sample in this study consisted of junior high school teachers in Indonesia ($n = 12$). We determined the participants through the purposive sampling method [55,56]. The criteria for participants were junior high school teachers with knowledge and experience in implementing differentiated learning and who were voluntarily willing to participate in this study. Choosing junior high school teachers as the sample was enacted as they teach junior high school students in Indonesia. Who are generally early adolescents with labile characteristics and need more attention. Therefore, we concluded that differentiated learning as a new teaching system in Indonesia is necessary to be researched.

To find the designated participants, we conducted a row of processes. First, we communicated with three principals of junior high schools in western Indonesia to submit research proposals and requests for research permits. In this process, the three principals act as officials who permit researchers to interview teachers in their respective schools. Besides that, the principal also provides information about which teachers meet the criteria to participate in this study. We obtained 17 names of teachers recommended to be participants from these school principals. Then we contacted the 17 teachers via email and WA (WhatsApp) messenger by including an explanation regarding the certificate of fulfillment of research ethics from a credible institution [57,58], the process and objectives of the research to be carried out, the rights and obligations of the participants during the research process, and the participant consent form. Of the 17 participants, 12 people agreed to become participants, while 5 of the 17 teachers refused for various personal reasons.

### 2.3. The Study Context

This research was conducted amid the educational system in Indonesia's current transformation. At present, the system is attempting to overcome the learning loss due to the COVID-19 pandemic. One of the programs offered by the Indonesian ministry of education and culture, research, and technology (*Kementerian Pendidikan, Kebudayaan, Riset, dan Teknologi*) is the *Pancasila Student Profile* development program through the application of the concept of differentiated learning. Some teachers in Indonesia have implemented differentiated instruction, but some have had difficulty despite attending training on differentiated instruction.

Therefore, the researchers tried to explore the experience of Indonesian teachers who have conducted differentiated learning to develop the Pancasila Student Profile through intracurricular activities. The research attempted to provide a picture of the model/pattern of how ideal differentiated learning is according to teachers in Indonesia. Thus, Indonesian teachers and foreign teachers with similar learning cultures can reflect the model of differentiated instruction we identified.

### 2.4. Study Instruments: Unstructured Interview Guide

The data collection process was carried out through in-depth individual interviews [59]. We applied an unstructured interview guide. Interview questions were framed into three main learning domains: planning, implementation, and assessment of differentiated instruction. In the interview process, researchers developed questions in accordance with the development of answers from participants; however, the developed questions were still within the same domain frame, namely about preparation, implementation, and the assessment of differentiated instruction. Therefore, certain interviewees face the same questions but may differ from others.

### 2.5. Interpretive Phenomenological Analysis

We analyzed the research via interpretative phenomenological analysis (IPA) [60]. The IPA explores how participants give meaning to their experiences, phenomenon, and certain conditions in their private and social lives. The data analysis process uses ATLAS.ti 9, qualitative data analysis and research software [61]. In general, we conducted data analysis through 6 stages, as shown in Figure 1 below.

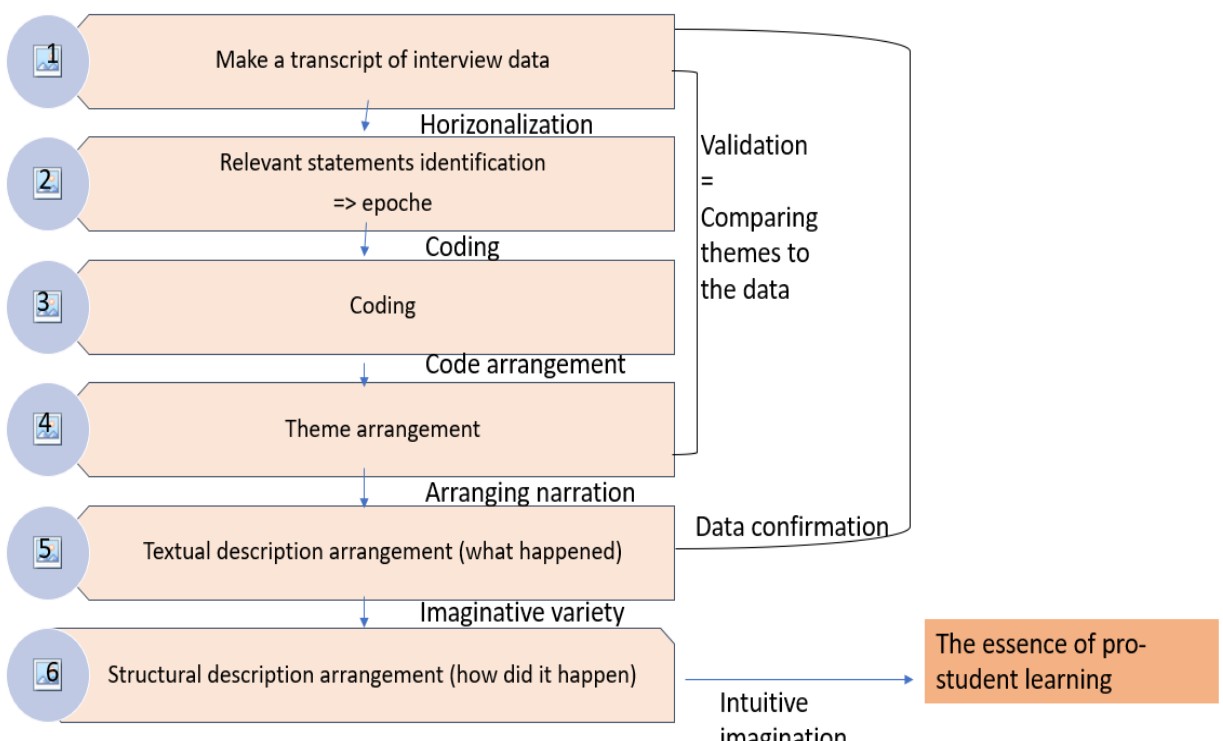

**Figure 1.** Data analysis process.

To be more specific about the data analysis process in Figure 1, we described the entire process. In more detail, the analysis conducted via ATLAS.ti with IPA is as follows:

### 2.5.1. Coding Process

The first step is to code every transcribed interview line-by-line [62]. This process is carried out by reading the entire analyzed transcripts, then finding the essential coding of each

participant's statement relevant to the research question. ATLAS.ti 9 was used to create a coding mark in every transcript. An example of the coding process can be seen in Figure 2.

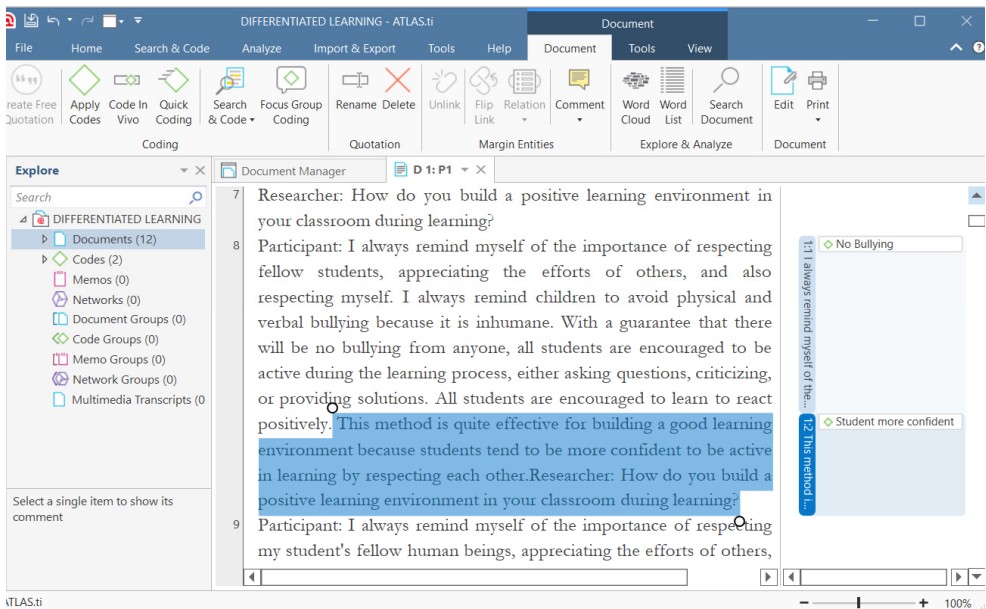

**Figure 2.** Coding process.

2.5.2. Research Result Formation

The next step is to group codes with similar meanings into one theme. From the generated 20 codes, We grouped them into four themes. The results of grouping into themes are shown in Figure 3.

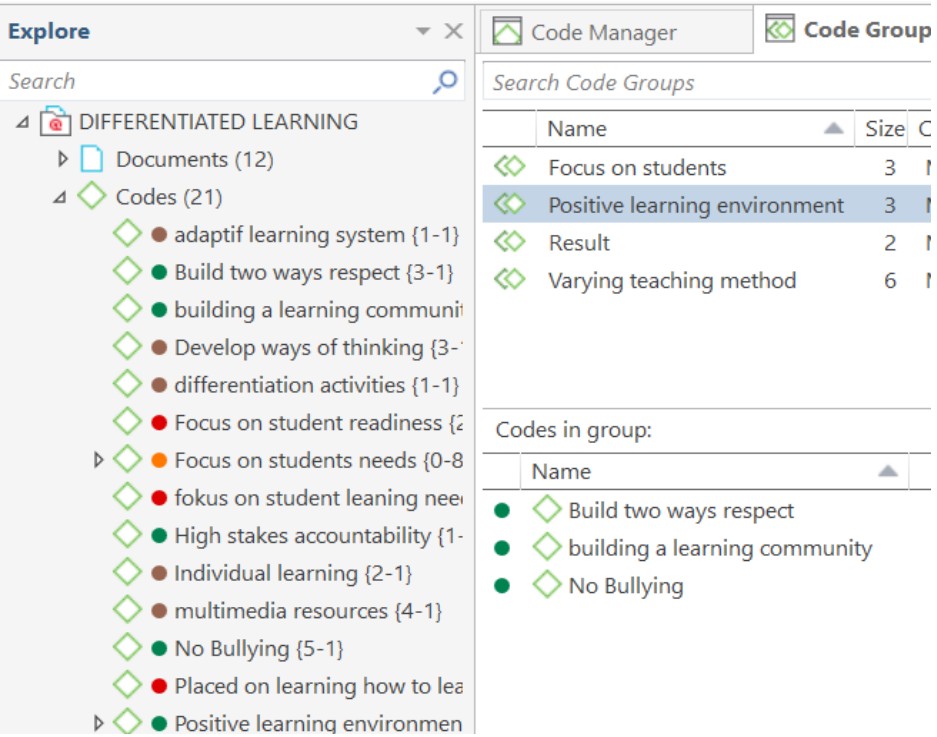

**Figure 3.** Theme construction process.

### 2.5.3. Textual Description Arrangement Formation (Textual Narrative)

After the four themes have been generated, the next step is to build a textual narrative to get an accurate picture of what happened. While compiling the narrative, we also guaranteed the correctness of the data by checking and comparing formed themes with the data in the form of relevant quotes from the participants.

### 2.5.4. Structural Description Formation

Next, we built a structural description, which entailed the development of a process of discovering the essence of research results from grouped data. We constructed the meaning of each theme using the background of positive educational theory [63] and educational psychology [64,65], mainly positive psychology in education. From the successfully constructed themes, two conceptual essences were found from the participants' experience of implementing differentiated learning: providing a sense of physical and emotional safety for students and students working on their respective roles. The conceptual model construction process in this study can be seen in Figure 4.

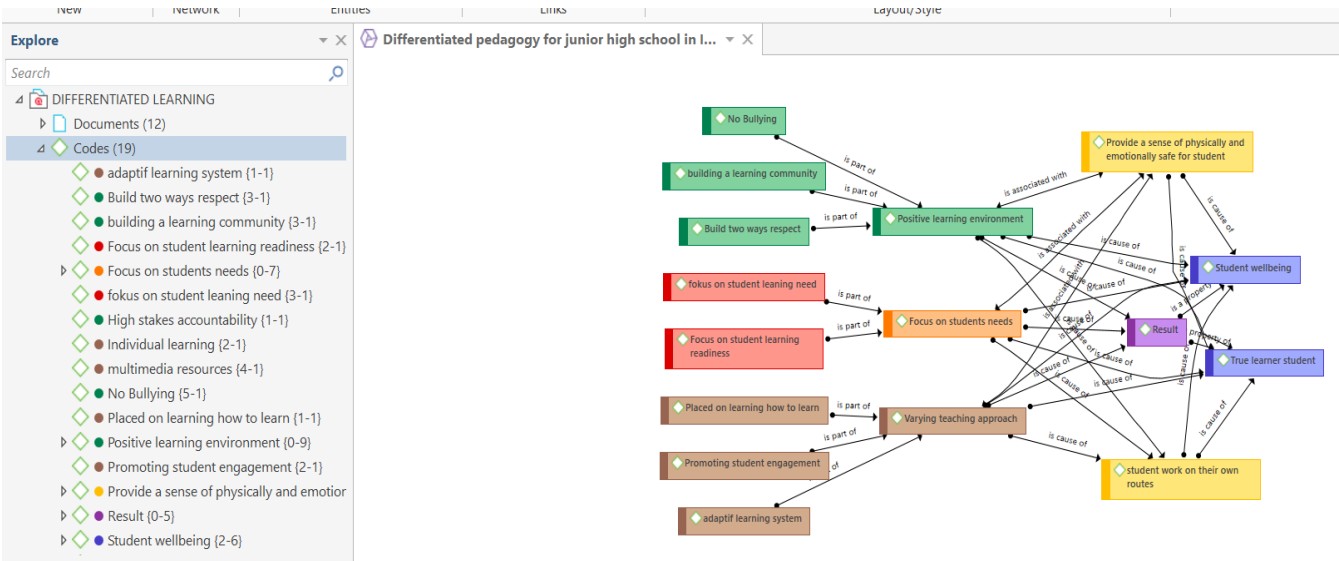

**Figure 4.** The development process of connection between concepts.

### 3. Results

This study reveals the experiences of junior high school teachers in Indonesia in implementing differentiated learning programs. From the participants' statements, themes can be constructed as the prominent domains representing the meaning of the differentiated learning program. According to the participants, three main domains are individually and collectively considered to be essential representative factors with particular dimensions of the concept of differentiated learning in Indonesia, including strategies for developing a positive learning environment, implementing varied learning strategies, and student-centered learning outcomes.

### 3.1. Developing a Positive Learning Environment

Participants stated that establishing a positive learning environment is one of their efforts to provide differentiated learning. From a teacher's experiences in classroom management, teachers strive to create a safe and bullying-free learning atmosphere for all class members, develop mutual respect, and build a learning community. These three things refer to a fundamental concept that differentiated learning can be developed by developing a positive learning environment. These three things can be explained in detail as follows:

### 3.1.1. No Bullying

According to the participants, a positive learning environment is a learning environment that can maintain and build a sense of security for students, both mentally and physically. The concrete form of a physically and mentally safe environment is an environment that is free from bullying, both bullying from teachers to students and between students. The following are some relevant statements regarding classes that nurture students' mental health:

> In my experience, students will be more cheerful and enthusiastic about learning when they get good treatment from the teacher and fellow students in class. This is a challenge for us as teachers to be able to build a sense of security and comfort physically and mentally so that students can learn well; the most obvious way is to ensure that there is no bullying in schools.
>
> (P1, lines 5–7)

Other participants explained the importance of a learning environment that can provide a sense of physical and mental security to achieve successful learning. The participants showed the best experience in the development pattern of a learning environment that can provide a sense of security as a form of teachers' favorability towards the differences in student potential. The following is an explanation from P4 regarding this:

> During the learning process, as a teacher, I always [emphasize] the importance of respecting fellow human beings, respecting one's efforts, and respecting yourself by avoiding bullying, physically and verbally. At the beginning of learning, children are always accustomed to making class agreements so that learning runs accordingly. One of the agreement's contents is not to bully anyone. With the guarantee that there will be no bullying from anyone, students feel more confident to be active in class during the learning process, either asking questions, criticizing, or even providing solutions. Students are also always directed to have a positive reaction to others; thus, the learning environment becomes a positive environment for children's development even though they have various conditions and different learning readiness
>
> (P2, lines 11–23).

The statement of P4 is in line with P9, which supports the importance of building a learning environment with a sense of security for students because it is closely related to students' mental health. In this case, P9 states that differentiated learning is a realization of the principle of learning that favors students so that to implement these principles, in the learning process, teachers think about the various impacts of the learning environment on the psychological development of children. The teachers try to develop a positive learning environment to make the children's mental growth healthy and optimal. P9 stated:

> At the beginning of every lesson, apart from praying, I always start learning with mutual agreement about class rules, learning procedures, and attitudes that need to be developed during the learning process. This method is quite effective in creating a conducive learning environment, especially in building mutual respect, self-confidence, discipline awareness, and anti-bullying. Based on my experience so far, making mutual agreements before learning is very effective in controlling student behavior and building a conducive classroom atmosphere without forcing the teacher's will
>
> (P9, lines 6–10).

P12 also believes that differentiated learning will be realized if all members of the class understand the existence of individual differences. Everyone should feel safe in class without physical or verbal bullying. Below is one of the relevant statements submitted by P12.

> Bullying is a major problem that often causes mental problems and a loss of confidence in its victims. So that all children have confidence in class and dare

to express their thoughts without being ashamed or afraid of anyone, I always remind them to keep away from verbal and physical bullying toward anyone

(P12, lines 7–11).

### 3.1.2. Building a Two-Way Respect

The results of the data analysis show that several keywords indicate the participants' views about a positive educational environment. The efforts to develop two-way respect, both inside and outside the classroom, are a form of teacher understanding and concern for the development of student character and the psychological development of students to grow positively. The participants said that the students were directed to build respect for others during the learning process even though the other parties were different from them. An example of the behavior mentioned by participant 1 (P1) is that respect can grow in a person when they can respect whomever he is talking to. Therefore, P1 continuously trains the classes to listen when someone else is talking in the classroom. In this case, P1 adds:

To build mutual respect between friends, I have a program called 'My Day' during distance learning. This program is one of the strategies that I do to raise the awareness of each child that everyone has a different life, so everyone must respect each other. The strategy I use to get students used to respecting other people is that in every first 10 min of class, I ask the children to take turns telling stories about themselves and their experiences. Other friends listen well, then they give a response via emoticons on the zoom layer or respond to it positively. Every time we meet, at least one child tells a story

(P1, lines 15–26).

P5, a *guru penggerak* (mover teacher), shared their experience regarding implementing differentiated learning that focuses on students' social and emotional development so that they have respect for others. The following is a relevant statement from P5 regarding this:

In implementing differentiated learning, I make an effort to grow students' respect for fellow human beings. Respect comes from the heart because there is empathy for others. So, in learning, I often do role-playing practices to foster empathy and respect among students. Indeed, it is not an easy thing and cannot be done once and immediately succeed. It takes patience to do it

(P5, lines 11–16).

### 3.1.3. Building a Learning Community

Several participants highlighted the importance of building a learning community to support the optimal development of student potential through developing a collaborative attitude and the development of networks among students. P5, for example, states that a positive learning environment is the first space for students to develop themselves both in terms of cognition and psychology. The existence of opportunities for students to be able to work collaboratively with their friends is the best way to build students' awareness that differences between students can be the best solution to solve various complex life problems. P5 emphasizes that the teacher is the manager and producer to create a successfully differentiated learning program setting, which focuses on developing the potential of each student through group activities.

Another participant (P6) also has a view that is in line with P5 that the pattern of working in partnership between students is one of the alternative learning programs that is oriented toward children's needs. The following is a direct statement from P6:

For the past two years, during the COVID-19 period, I have always tried to have a group learning process, even if only through Zoom meetings or video calls. I ask the children to hold regular meetings between group members to do the tasks together. According to the reports that the children gave, during independent group work, the students carried out peer mentoring and peered

lessons. Students who are good at guiding students who have not succeeded in mastering the teaching material. These activities help children learn more easily, eliminate the feeling of loneliness, and make it easier for me as a teacher because the peer guidance process in each study group makes learning easier for students to understand

(P6, lines 17–25).

P11, a junior high school social studies teacher, said that one of the ways they do differentiated learning is to build a positive learning environment. According to P11's explanation, the main principle of differentiated learning is respecting differences. For students to appreciate differences, they need to be accustomed to getting along and working with others. For this reason, P11 formed study groups between students in the class and students from other classes, even with students from other schools.

Even though during the COVID-19 pandemic, learning was carried out online, I still encouraged children to form learning communities, whether with classmates or with friends from different classes. For grade 9, they have also succeeded in forming a learning community with fellow grade 9 students from other schools. I do it by working with fellow teachers in other schools to connect students to get to know each other and then form inter-school social studies learning community. From the responses and reflections made by the children, they are happy to join this learning community program

(P11, lines 15–22)

### *3.2. Varying Teaching Methods*

The teaching strategies were the second domain that became the participants' focus regarding the differentiated learning program. The participants stated several things that can be identified as differentiated forms of teaching. Namely, teachers set teaching patterns to meet the learning needs of students with different potentials. According to the participants, teachers can ideally do things to stimulate interaction between students and teachers, open up opportunities for students to make decisions in learning, and provide opportunities for students to be more creative and innovative. In more detail, the following are excerpts from relevant statements from participants that show these data.

### 3.2.1. Promoting Student Engagement

The results of data analysis show that the differentiated learning process is a learning program intended to equip students with various skills and character values to adapt to various events in their lives according to their natural potential. For this reason, the participants shared that one way to sharpen the character and skills of students effectively is to involve students in an intensive learning process.

One way to increase student involvement in learning is through project-based learning. The following is P4's explanation of his experience in teaching junior high school students during the COVID-19 period who succeeded in increasing student involvement in learning:

Schools were closed during the COVID-19 period, and learning was carried out from home online. So that the children do not feel alone, I always set learning in the form of project-based group work. Since the first meeting, I have given the project learning steps every week. In the student worksheets, I set them so that individually, they report what was done in their group. So even though they work as a group, they individually have their own responsibilities.

Another participant, P8, had a similar experience to P4. In an effort to increase student involvement during learning, there are particular tasks that P8 gives to students. The following is the presentation of P8:

Oh . . . yes, in the context of classroom learning, I realize that students will absorb teaching materials more quickly if they are actively involved in learning,

starting from concept building or in the form of applications. Well, based on my experience in teaching, the things that students like are when at the beginning of the lesson, the teacher explains in advance what is being assessed, how the learning steps are carried out, and the work is done in groups

(P8, lines 45–49).

### 3.2.2. Placed on Learning How to Learn

The first step that a teacher needs to take is to develop the right mindset for students to become better learners. Developing students' mindsets help build the belief that they can become anything they want to be with enough effort. Below are some statements made by teachers to build a growth mindset among students, as stated by P6:

I often show children various motivational videos, especially videos that can provide understanding and awareness of how our minds learn new concepts biologically and demonstrate strategies for effective learning

(P6, lines 67–69).

P7 also stated that one of the right ways to equip students to have the ability to learn is to raise awareness about the importance of learning and find out how to learn that is suitable for them. According to P7, children need to be accustomed to reading a lot. So learning is set so that children are accustomed to finding the essence of a phenomenon, expressing opinions, and being able to solve problems. The teacher can set this habit through a lesson plan.

### 3.2.3. Adaptive Learning System

According to the participants, the main challenge faced by teachers in Indonesia today is that teachers and teaching staff have difficulty monitoring student performance one by one in-depth. Especially when learning is carried out through distanced learning methods, such as during the COVID-19 period, communication that occurs on virtual platforms is minimal, and the majority of it goes one way. Therefore, teachers have limitations in providing different subject matter according to students' abilities. To overcome this, the participants have the experience to overcome it by implementing an adaptive learning system so that students can independently develop themselves. The statement of P10 is as follows:

In a condition like this, yes, ma'am, I have difficulty controlling whether the students have mastered the learning competencies. So to overcome this, I tried to adopt various free learning apps. These apps are handy and flexible. I can control students whenever they come to class, work, and do what they do individually

(P10, lines 71–74).

The participants acknowledged that the adaptive learning system is good for carrying out personalized learning even though it is still in the experimental stage and its implementation is gradual. The following is P9's statement regarding this matter:

To provide educational services that follow the potential of each student, I try to implement a different instructional system for each student according to the readiness of each student to learn. This is still an experimental process, ma'am... I use the diagnostic assessment results to develop a range of instructional instruction suitable for individual students. I did this process with the help of a simple application that I created based on excel. In the future, I plan to develop this adaptive learning system more professionally so that learning is more effective and efficient

(P9, lines 67–75).

### 3.3. Learning Outcomes That Focus on Individual Students

The third aspect that became the participants' focus regarding the concept of differentiated learning was related to learning outcomes, which emphasized developing the skills

and feelings of individual students. The participants stated that understanding teaching materials in the form of a series of knowledge is not the main goal of the learning process. They are merely a tool to shape the character and competence of students to become true learners and achieve student well-being.

### 3.3.1. True Learner Student

According to the participants, learning outcomes should not just add specific knowledge to the curriculum but also grow the character of true learners. The parts of knowledge in the curriculum are only mediating tools to help children and abilities expected to grow and be embedded in the students. The following is P3's statement regarding this matter:

> The ideal learning outcomes are outcomes that are beneficial to the lives of students and others. So, when teaching, I always try to instill the attitude of a true learner. I often tell children that the best way to become a true learner is to learn to listen to other people talk even though you know it already. From what we hear from other people, at least we can learn something new or something different from before. Even when you listen to other people's wrong statements, you will know how so you don't make the same mistake in understanding something

> (P3, lines 58–63).

A statement made by P7 supports P3's statement:

> I believe that students need the ability to become true learners, not only as long as they are students but also to equip them for life. They need to be educated with adaptive skills and abilities. A true learner is someone who can see learning opportunities and then differentiate effectively to focus only on those that will add value to their knowledge and development, as is needed today

> (P7, lines 56–60).

### 3.3.2. Student Well-Being

Achievement of student well-being [66] is the end of a learning process. The concept of well-being, according to the driving teachers, is a holistic concept, both from the maturity of knowledge, the fulfillment of needs, and the desire to feel loved.

P4 stated that the ideal educational outcome's final estuary is the emergence of pride and happiness in students because they feel their learning needs are being met. The elements that make up students' well-being are the teacher's attention, two-way communication, and the opportunity for children to express their feelings. The following is the relevant statement submitted by P4:

> Whenever I teach, I often ask the children what they need, what they want to do, and how they feel. Students tend to be more open when we listen to them well and provide feedback on what they have to say. Children give a signal that when wishes come true, they can feel proud and happy. Well . . . a teacher's job is to nurture children to have positive expectations so that student well-being becomes genuine well-being, not a momentary pleasure

> (P4, lines 89–94).

## 4. Discussions

The themes that emerged from the results of the participants' experience constructs regarding differentiated learning, if visualized, look like in the following Figure 5.

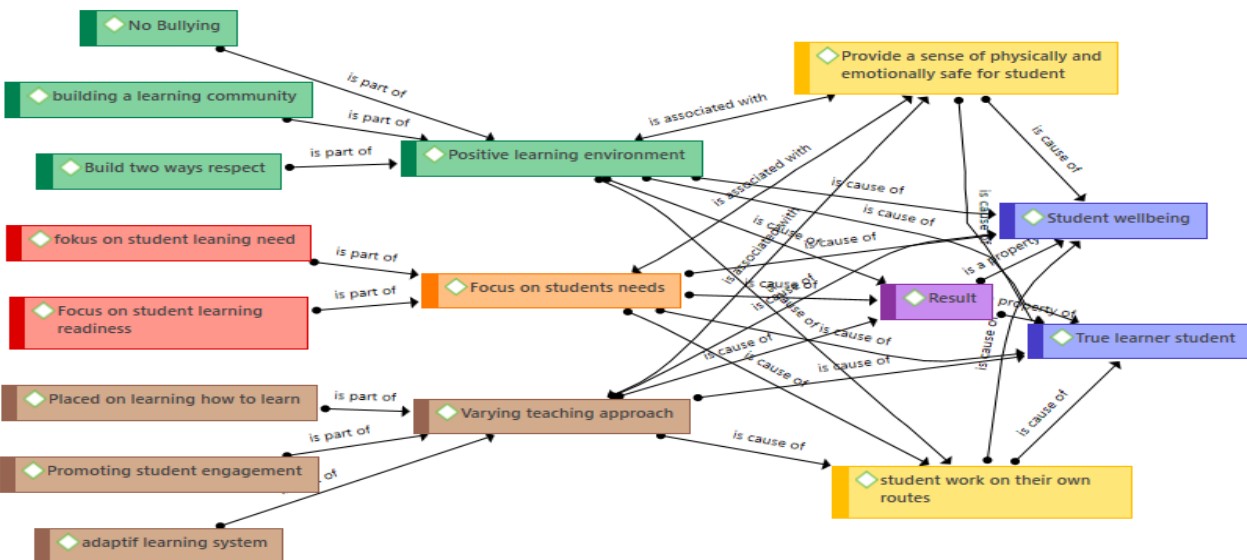

**Figure 5.** Conceptual model of differentiated learning in Indonesia.

Based on Figure 5, the implementation of differentiated learning in the participants' classes is carried out by teachers who understand and are concerned for students' physical and emotional needs. The teachers try to guarantee the safety and comfort of students during the learning process. In addition, various teacher behaviors in the classroom show that differentiated learning enables children to work in harmony with their world. In more detail, this phenomenon can be explained as follows:

### 4.1. Teachers Provide a Sense of Physically and Emotionally Safe for Student

Differentiated learning was developed by the participants in this study, applying pro-student learning practices. This matter can be seen in learning practices that apply various learning methods according to student learning styles, developing a learning environment that supports the achievement of personal learning outcomes, and paying attention to student needs during the learning process [67]. The teachers' point of view, which prioritizes the achievement of learning outcomes in the form of student well-being and true learner, becomes an indicator that the differentiated learning implemented by the participants is realizing the philosophy of progressive education [68] in which students have a role as a subject in the learning process. Teachers develop learning methodologies and evidence with clear success criteria for each student, which is strong evidence that teachers are trying to implement and assess learning effectiveness [69] in favor of students [70]. What was carried out by the participants was in line with the principle of transparent assessment, where the indicators of achievement of learning outcomes are adjusted to the ideal picture of graduates in the national education system [71–73] without compromising the rights of students who have different potentials and different learning needs.

The concept of differentiated learning applied by the participants makes the potential and needs of students the fulcrum of learning development [74], which encourages students to be active in learning [75]. This matter is in line with Han et al.'s research, which states that effective learning is learning that can connect the learning process with the needs of students [76].

Viewed from the point of view of educational psychology, the development of a positive learning environment is proof that teachers are concerned about maintaining students' mental health [77]. The declaration of a school without bullying, the development of two-way respect, and the development of a learning community are evidence that the participants are trying to carry out learning that builds students' awareness to coexist with one another in diversity [78].



In this context, the essence of differentiated instruction that possesses the potential to be developed in the Pancasila Student Profile is how teachers should provide a sense of physical and emotional safety for students. The safety is aimed to grow a comfortable feeling of students and protect them from a negative perception of teachers' justice [79], whether they have different potentials and learning styles.

*4.2. Students Work on Their Respective Routes*

The results showed that teachers understand the concept of differentiated learning as learning that seeks to place students as learning subjects. The teachers position themselves as a facilitator in learning who seeks to provide a learning environment that follows the needs of students, a child-centered learning process, and learning outcomes that emphasize the development of students' potential to become true learners [80]. The teachers seek to provide broad opportunities for students to take advantage of various learning resources and choose various learning activities according to each student's interests and natural talents [81]. This matter is in line with Ki Hadjar Dewantara's thoughts [82] that education is a process of guiding, not demanding.

The learning environment is an essential aspect in order to build pro-student learning. The learning environment in the participants' spotlight is not just the physical environment; it emphasizes the school's cultural environment, which can have a positive social and emotional effect [83], wherein realization requires support from all parties. An environment that can provide a sense of security physically and mentally [84,85], develop a culture of mutual respect [86,87], and build a learning community [8,88], is an environment that is identified as a positive and pro-student environment. This idea supports the results of previous research, which states that the learning environment in schools has a vital role in student development. In particular, the differential effects found for different educational pathways highlight the educators' awareness of individual differences between their students [89].

Various learning approaches [90] and putting students as subjects are believed to build students' learning abilities according to their respective routes. In other words, this research shows that a differentiated learning program is a learning program that tries to touch students' lives directly and motivate students to be actively involved in discovering new things. The learning approach that pays more focuses on students' physics and mentality (Malafantis, 2021) is proven to increase students' ability to learn independently and understand the meaning of each learning experience they do [91,92]. This matter is in line with the values developed in the Pancasila Student Profile, i.e., Indonesian learners are lifelong learners who are competent, possess character, and behave according to the values of Pancasila [27]. When students are capable of learning independently according to their learning styles and needs, their learning experiences can support achieving student well-being [93].

**5. Conclusions**

In this study, we aimed to contribute to the increasing insights into differentiated learning practices by junior high school teachers in Indonesia. The context of the Indonesian environment, which is filled with Asian cultural backgrounds and various values and cultures, was the starting point for developing understanding and discussion.

This study notes that the differentiated learning process is learning that can provide real benefits for everyone involved, especially in developing learning abilities and student sense of personal well-being. To produce effective learning, teachers need to develop insight and awareness about the existence of diversity that also needs space to develop into its natural potential. Our analysis produces a conceptual model of differentiated learning that is meaningful to students by applying a combination of progressive insights and teacher skills in adapting to changing contextual learning systems. Differentiated learning is learning that relies on efforts to develop student competencies, not on the transfer of

teaching materials, to produce the ideal quality of graduates as outlined in the concept of the Pancasila Student Profile.

**Author Contributions:** Conceptualization, E.H.; methodology, E.H.; software, M.I.A.B.; validation, S.S.; formal analysis, E.H.; investigation, S.S., I.M. and M.I.A.B.; resources, S.S., I.M. and Y.F.; data curation, E.H.; writing—original draft preparation, E.H.; writing—review and editing, E.H. and M.I.A.B.; visualization, S.S., I.M. and M.I.A.B.; supervision, I.M. and Y.F.; project administration, I.M. and L.P.; funding acquisition, E.H. All authors have read and agreed to the published version of the manuscript.

**Funding:** This research was funded by the Directorate of Research and Community Service (Direktorat Riset, Teknologi, dan Pengabdian Kepada Masyarakat/DRTPM) of the Indonesian Ministry of Education and Culture in the form of a Higher Education Excellence Basic Research Grant (Penelitian Dasar Unggulan Perguruan Tinggi/PDUPT) grant number 036/E5/PG.02.00/2022. And The APC was funded by DRTPM grant number 028/PB.PDUPT/BRIn.LPPM/VI/2022, Information regarding the funder and the funding number should be provided.

**Institutional Review Board Statement:** The study was conducted in accordance with the Declaration of Helsinki, and approved by the Institutional Review Board (or Ethics Committee) of KOMITE ETIK PENELITIAN (KEP UAD) (protocol code 012208106, Date approval 12 May 2022).

**Informed Consent Statement:** Written informed consent has been obtained from the patient(s) to publish this paper.

**Data Availability Statement:** Not applicable.

**Conflicts of Interest:** The authors declare no conflict of interest.

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
