# Peer review of "Conceptual Model of Differentiated-Instruction (DI) Based on Teachers’ Experiences in Indonesia"

_education, doi:10.3390/educsci12100650_

Round 1
Reviewer 1 Report
Conceptual Model of Differentiated Learning Based on Indonesia Teacher's Experiences
Thank you for the opportunity to review this manuscript.
With a view to improving the output, do consider the following:
Create a more explicit link between Pancasila (Student Profiles) and teacher concerns.
The lit review requires more external support (especially for broad statements like DI being linked to socio-emotional awareness). This section should be mentioning the findings of actual empirical studies. Given the direction of the study, it would be good to cite other empirical studies which link Differentiated Instruction and Social and Emotional positions.
The study is qualitative in nature and utilises phenomenology – more detail should be provided about the appropriateness of this form of analysis/method for the project at hand.
There was demonstrated rigour in the Methodology but this section requires better organisation and structure. Delineate Sampling, Analysis and Procedures separately. This section is also completely without citations. The varying steps undertaken to facilitate the study should be supported by other research. The steps should also be clearly delineated so that they can be replicated in future studies.
Link the Results of the study more explicitly to Pancasila.
More synthesis is required in the Discussion – for example, align like studies by drawing specifically on the findings of other studies.
The manuscript should be thoroughly proofread to reduce language inconsistencies.
The reference list is impressive.
Author Response
First and foremost, we express our sincere appreciation for allowing us to revise further the manuscript (Manuscript ID: education-1788971) entitled "Conceptual Model of Differentiated Learning Based on Indonesia Teacher's Experiences" for continued consideration publication in Education Sciences.
Attached you will find a revised paper for your review. We have carefully read and considered the comments and suggestions you and the two thoughtful reviewers provided. Below, we respond to or detail how we have addressed the various concerns raised.
Concerns raised by Reviewer I:
To improve the output, do consider the following:
- Create a more explicit link between Pancasila (Student Profiles) and teacher concerns.
Thank you for noting this issue; with the balanced and helpful advice given by the reviewers, we have modified the introduction part, and the "propose of the study" section to organize the information logically to create a more explicit link between Pancasila (Student Profiles) and teacher concerns in page 3, lines 58 to 70.
- The lit review requires more external support (especially for broad statements like DI being linked to socio-emotional awareness). This section should mention the findings of actual empirical studies. Given the direction of the study, it would be good to cite other empirical studies which link Differentiated Instruction and Social and Emotional positions.
Thanks for this constructive suggestion. We add to previous research confirming the relationship between DI and students' socioemotional on pages 3, lines 85-89.
- The study is qualitative in nature and utilises phenomenology – more detail should be provided about the appropriateness of this form of analysis/method for the project at hand. There was demonstrated rigour in the Methodology but this section requires better organisation and structure. Delineate Sampling, Analysis and Procedures separately. This section is also completely without citations. The varying steps undertaken to facilitate the study should be supported by other research. The steps should also be clearly delineated so that they can be replicated in future studies.
We thank the reviewer for raising this importance in the method and materials section. As both reviewers stated, the authors should provide additional details in the method section, including, among other things, how they determined the present sample size. This part has been modified with the correct information (see pages 4-5, methods and materials).
- Link the Results of the study more explicitly to Pancasila. More synthesis is required in the Discussion – for example, align like studies by drawing specifically on the findings of other studies. The manuscript should be thoroughly proofread to reduce language inconsistencies.
We thank the reviewer for raising this manuscript. The section remarked has been modified, adding significant information explained in the present study. At the same time, cohesion and language have been checked.
- The reference list is impressive.
Thank you very much.

Reviewer 2 Report
The paper is of some value to the potential international audience. However, there can be further refinements and elaborations about why developing a differentiated learning model based in Indonesian teachers' perspectives are important to us as international readers. In other words, what this model can contribute to the current literatures on differentiated instruction and how this study can address the research gap should be further established.
Moreover, in the current study, the targeted participants are junior high school teachers. But why only junior high school teachers are selected? What are their situations in Indonesia? This is quite vague to understand the study without knowing the relevant socio-cultural backgrounds of what's going on in context.
On the other hand, there is a strong need to clarify the Section on "Materials and Methods" as the Author(s) mentioned "The Materials and Methods should be described with sufficient details to allow others to replicate and build on the published results. Please note that the publication of your manuscript implicates that you must make all materials, data, computer code ... ", in which the contents are confusing and not directly relevant to the objective of the paper.
Apart from the above, there should be a clearer presentation of why phenomenological research is adopted in the study and what kind of phenomenological research is selected - descriptive or interpretivist? There should be further discussion on the choice of research approach in relation to the current study's purposes.
Concerning data collection and analysis, there has to be an illustration about how this phenomenological study can ensure trustworthiness and what ethical considerations have been taken into account.
Author Response
First and foremost, we express our sincere appreciation for allowing us to revise further the manuscript (Manuscript ID: education-1788971) entitled "Conceptual Model of Differentiated Learning Based on Indonesia Teacher's Experiences" for continued consideration publication in Science Education.
Attached you will find a revised paper for your review. We have carefully read and considered the comments and suggestions you and the two thoughtful reviewers provided. Below, we respond to or detail how we have addressed the various concerns raised.
Concerns raised by Reviewer II:
To improve the output, do consider the following:
- The paper is of some value to the potential international audience. However, there can be further refinements and elaborations about why developing a differentiated learning model based in Indonesian teachers' perspectives are important to us as international readers. In other words, what this model can contribute to the current literatures on differentiated instruction and how this study can address the research gap should be further established.
Thank you for noting this issue; with the balanced and helpful advice given by the reviewers, we have modified the introduction part, and the "reason of the study" section to organize the information logically for international audience about DI in Indonesia as multiculture country in page 3, lines 58 to 70.
- Moreover, in the current study, the targeted participants are junior high school teachers. But why only junior high school teachers are selected? What are their situations in Indonesia? This is quite vague to understand the study without knowing the relevant socio-cultural backgrounds of what's going on in context.
Thanks for this constructive suggestion. We added an explanation of why we chose only junior high school teachers on page 3, lines 131-134
- On the other hand, there is a strong need to clarify the Section on "Materials and Methods" as the Author(s) mentioned "The Materials and Methods should be described with sufficient details to allow others to replicate and build on the published results. Please note that the publication of your manuscript implicates that you must make all materials, data, computer code ... ", in which the contents are confusing and not directly relevant to the objective of the paper.
Apart from the above, there should be a clearer presentation of why phenomenological research is adopted in the study and what kind of phenomenological research is selected - descriptive or interpretivist? There should be further discussion on the choice of research approach in relation to the current study's purposes.
Concerning data collection and analysis, there has to be an illustration about how this phenomenological study can ensure trustworthiness and what ethical considerations have been taken into account.
We thank the reviewer for raising this importance in the method and materials section. As both reviewers stated, the authors should provide additional details in the method section, including, among other things, how they determined the present sample size. This part has been modified with the correct information (see pages 4-5, methods and materials).
